# Spatial and Temporal Differences in the Green Efficiency of Water Resources in the Yangtze River Economic Belt and Their Influencing Factors

**DOI:** 10.3390/ijerph18063101

**Published:** 2021-03-17

**Authors:** Chong Huang, Kedong Yin, Zhe Liu, Tonggang Cao

**Affiliations:** 1School of Economics, Ocean University of China, Qingdao 266100, China; huangchong@stu.ouc.edu.cn (C.H.); sdlwsge@stu.ouc.edu.cn (Z.L.); 2Institute of Marine Economy and Management, Shandong University of Finance and Economics, Jinan 250014, China; yinkedong@ouc.edu.cn; 3School of Management Science and Engineering, Shandong University of Finance and Economics, Jinan 250014, China; 4Ocean Development Research Institute, Major Research Base of Humanities and Social Sciences of Ministry of Education, Ocean University of China, Qingdao 266100, China; 5School of Business and Management, Queen Mary University of London, Mile End Road, London E1 4NS, UK; 6College of Environment Science and Engineering, Ocean University of China, Qingdao 266100, China

**Keywords:** Yangtze River Economic Belt, green efficiency of water resources, SBM-DEA model, Malmquist index, social network analysis, system GMM model

## Abstract

Using panel data from 11 regions (9 provinces and two cities) in the Yangtze River Economic Belt (YREB) during 2002–2017, the regional differences in and spatial characteristics of the green efficiency of water resources along the YREB were analyzed. The undesirable outputs slacks-based measure-data envelopment analysis, Malmquist index, and social network analysis models were employed. A dynamic panel using a system generalized method of moments model was established to empirically examine the main factors influencing green efficiency. The results show the following. First, temporally, green efficiency fluctuates while showing an overall decreasing trend; spatially, green efficiency generally decreases in this order: downstream, upstream, then midstream. Second, the change in the total factor productivity (TFP) index shows an overall increasing trend, with TFP improvement mainly attributable to technology. Third, green efficiency shows a significant spatial correlation. All provinces are in the spatial correlation network, and the network, as a whole, has strong stability. Finally, water resource endowment, water prices, government environmental control strength, and the water resources utilization structure have a significant impact on green efficiency.

## 1. Introduction

The Yangtze River Economic Belt (YREB), as the world’s largest river basin economic belt, is an important new engine for China’s economic development in the “new normal.” This belt spans the three major zones of East and West China, covering nine provinces and two cities (hereafter, “11 YREB regions”): Shanghai, Jiangsu, Zhejiang, Anhui, Jiangxi, Hubei, Hunan, Chongqing, Sichuan, Yunnan, and Guizhou. The YREB hosts over 40% of China’s total population and occupies 21.4% of the national territory. According to their geographical distribution along the Yangtze River Basin, the 11 YREB regions can be divided into upstream, midstream, and downstream. Water resources are among the core elements for ecological civilization construction and green development in the YREB. Analyzing the green efficiency of water resources (alternatively, “water resource green efficiency”) in the YREB is important for understanding the deep-seated laws of water resource utilization in the process of regional socioeconomic development and for realizing the optimal allocation of water resources and coordinated regional development in the YREB.

As an essential factor in production and life, water resource utilization efficiency has been a hot topic in academia. Improving water use efficiency is the primary way to sustain water resources [1,2]. An efficient and accurate evaluation method depends on water resource utilization and efficiency evaluation. This concern has been voiced by several studies [3,4]. Research on water resource efficiency has mainly focused on efficiency measurement. The common efficiency measurement methods include the index system evaluation [5], data envelopment analysis (DEA) [6], and stochastic frontier analysis (SFA) [7]; this study focuses on DEA, which is a systemic analysis method developed by Charnes et al. [8] based on “relative efficiency evaluation.” DEA has many advantages compared to other measurement methods: no prior determination of functional relationships, non-subjective weighting, and the possibility of analyzing the ineffectiveness of the decision-making unit. It is particularly effective for evaluating multi-input and multi-output decision making units’ efficiency. It has become a primary technical tool for assessing relative efficiency [9,10]. For example, Yang and Li [11] adopted a DEA model to calculate and evaluate the total factor efficiency of waste gas control in different industries in China. Zhang et al. [12] used the Malmquist index decomposition method to analyze green-biased technical changes in terms of industrial water resources in the YREB. Other scholars have used the DEA method to measure water use efficiency in different countries [13,14,15].

Most of the earlier water-use efficiency studies treated economic efficiency, that is, gross domestic product (GDP), as the only output [16,17]. However, this contrasts with the actual production process driving social development. Any calculation of economic efficiency by ignoring the cost of environmental pollution is both inaccurate and incomplete. For example, green water use efficiency also considers undesirable outputs [18,19,20]. The existence of undesirable outputs significantly impacts the efficiency assessment. Undesirable outputs such as water pollutants must be reduced as much as possible to achieve the best water-use efficiency [21,22]. Zhao et al. [23], Sun et al. [24], Ma, Ding, et al. [25], and Ma, Wang, et al. [26] explored the driving factors and influencing mechanisms of water resource utilization efficiency. They included pollutants as undesirable outputs in their evaluation of water resource utilization efficiency. Deng et al. [27] used the slacks-based measure (SBM) model to evaluate water resource efficiency in 31 provinces in mainland China. They proposed several factors affecting water use efficiency, including value-added agriculture, per capita water consumption, the output ratio of sewage units, and import/export dependence.

Guo et al. [28] used the SBM-Tobit method to calculate the green efficiencies of water resources in 18 cities of Henan Province from 2011 to 2018. This is a new method for the scientific measurement of water resources and green benefit evaluation. Mai et al. [29] applied the total factor productivity (TFP) framework to comprehensively assess China’s water use efficiency, considering environmental factors. They found apparent differences in water use efficiency between the eastern, central, and western regions. Xiao et al. [30] measured the green water resource efficiency of Chinese provincial industries by establishing an epsilon-based measure (EBM) model that considers undesirable outputs. Gao et al. [31] analyzed the green efficiency of air in the Chengdu–Chongqing region in China. Li et al. [32] used network DEA to estimate the eco-efficiencies of China at the provincial level.

Gong et al. [33], Zhang et al. [34], Pan et al. [35], and Wang and Xie [36] studied water resource green efficiency. They considered the impact of environmental regulations on water resource utilization efficiency and analyzed the spatial and temporal evolution and influencing factors of water resource green efficiency. Yang and Xie [37], Yu and Liu [38], and Lu and Xu [39] conducted static and dynamic analytical studies on water resource utilization efficiency using the Malmquist index to explore the factors influencing water resource green efficiency. Tian et al. [40] studied the water resource green efficiency of Chinese provincial industries and analyzed its regional differences and influencing factors. They also conducted an in-depth analysis of the factors influencing water resources from the perspective of input-output, water rights trading, and pilot markets.

Existing studies outline the theory and models of water resource green efficiency; however, they have some limitations. First, in studies on water resource efficiency evaluation, the measurement index is relatively simple. Second, there is no comprehensive and objective measurement of the regional water resource green efficiencies’ relationship. This study makes the following improvements based on the existing research: (1) it incorporates China’s human development index into the expected output evaluation index system; (2) it considers water resource green efficiency; and (3) it applies the improved index system to the study of water resource efficiency based on previous research. Using the SBM-DEA model, we measure the water resource green efficiency in the 11 YREB regions from 2002 to 2017. To undertake network analysis, we use the vector autoregression (VAR) Granger causality test to transform the “attribute data” of water resource green efficiency into “relational data.” Finally, we use the dynamic panel system generalized method of moments (GMM) model to analyze the factors affecting green efficiency.

The remainder of this study is organized as follows. Section 2 illustrates the study area, variables, data sources, and measurement methods. Section 3 presents the empirical results of the SBM-DEA, Malmquist index, social network analysis, and dynamic panel system GMM models, including the measures of and factors affecting the green efficiency of water resources in the YREB. The conclusions and recommendations are presented in Section 4.

## 2. Materials and Methods

### 2.1. Study Area and Data Sources

#### 2.1.1. Study Area

The YREB covers 11 provinces and cities. The upstream includes Chongqing, Sichuan, Guizhou, and Yunnan; the midstream includes Anhui, Jiangxi, Hubei, and Hunan; and the downstream includes Shanghai, Jiangsu, and Zhejiang. The belt covers an area of approximately 2,052,300 square km, accounting for 21.4% of the country’s total land. The population is approximately 600 million, which is more than 40% of China’s total population. The geographic position of the YREB is shown in Figure 1.

#### 2.1.2. Data Sources

Data are for the 11 YREB regions from 2002 to 2017. The required indicators are divided into two categories: input- and output-based. We selected water resources, labor, and capital as the input indicators; real GDP and the China Human Development Index (CHDI) as desirable outputs; and total wastewater discharge and chemical oxygen demand (COD) emissions from wastewater as undesirable outputs [41]. The specific indicators are as follows.

Water resource input: In regional economic growth studies, this indicator is generally the total amount of water used, that is, the sum of industrial, agricultural, domestic, and ecological water. The relevant data were obtained from the China Statistical Yearbook [42] and the China Water Resources Bulletin [43].Labor force input: This is the total number of persons employed at the end of the year in each province (urban and rural). Missing data for some years in individual provinces were derived from other years, based on the smoothing index. The relevant data were obtained from the China Statistical Yearbook [42] and the China Population and Employment Statistical Yearbook [44].Capital input: We adopt the “perpetual inventory method,” as proposed by Zhang et al. [45]. The traditional way of accounting for capital stock is to use fixed capital in previous years and deduct consumption from the total. The perpetual inventory method is based on new investments (total fixed asset formation). Considering the economic depreciation of capital stock in previous years, this method integrates capital stock accounting with capital services accounting. The traditional method accounts for capital stock alone. Capital stock was calculated for each province from the China Statistical Yearbook [42].Desirable output: The desirable outputs were the real GDP and CHDI for each province and city.

Real GDP: The corresponding deflator was applied to the regional GDP index (2002 = 1) to eliminate the effect of prices. Relevant data were obtained from the China Statistical Yearbook [42].

CHDI: With the introduction of the concept of “green development,” any study of water resource efficiency should consider that the economy should meet today’s social development requirements. The Fifth Plenary Session of the 18th Party Central Committee conceptualizes green development as comprising the five major development concepts of “innovation, coordination, greenness, openness, and sharing” [46]. In the context of these concepts, this study categorizes CHDI as the nation’s development needs along five dimensions: life expectancy, income, education, livelihood, and sustainable development (shown in Table 1).

Formula:

(1) Life expectancy index

The life expectancy index is measured by life expectancy at birth, published by the Department of Statistics for the entire country and for each province, for 1990, 2000, and 2010. For the remaining years, this value was obtained by linear interpolation.
Life expectancy index = (average life expectancy − 20)/(85 − 20),
where 20 and 85 are the thresholds for life expectancy.

(2) Education index

The education index is measured by average years of schooling.
Average years of schooling = (number of people in primary school × 6 + number of people in lower secondary school × 9 + number of people in upper secondary school × 12 + number of people in tertiary education and above * 16)/total number of people surveyed.
Education index = (average years of schooling − 0)/(15 − 0), 
where 0 and 15 are the thresholds for average years of schooling.

(3) Income index

The income index is measured using purchasing power parity (PPP) gross national income (GNI) per capita in US dollars. Data on China’s GNI per capita were obtained from the annual data published by the National Bureau of Statistics.
Income index = [log (GNI (ppp$) per capita) − log(100)]/[log (75,000) − log(100)], 
where 100 and 75,000 are the per capita income thresholds.

(4) Livelihood improvement index

The people’s livelihood index is the geometric average of the social security index and the Engel index. The Engel index was standardized from the Engel coefficient data. The social security index was standardized from the data on per capita social security expenditures.
Engel index = (0.8 − Engel coefficient)/(0.8 − 0.2), 
with 0.2 and 0.8 being the Engel coefficient thresholds.
Social security index = (social security level − 0)/(0.3511 − 0), 
and
Social security level = total social security expenditure/total GDP, 
where 0 and 0.3511 are the threshold values of the social security level; 0.3511 is the maximum value among provinces and municipalities in 2020, based on the 13th Five-Year Plan’s target value.

(5) Sustainable development index

The sustainable development index is the geometric average of the innovation development index, green development index, and openness index. The underlying data were obtained from annual data published by the National Statistical Office.
STI index = volume of research and development (R&D) inputs/total GDP.

The green development index is expressed in terms of carbon intensity, where
carbon intensity = total CO_2_ emissions/total GDP.
Openness index = total exports and imports/total GDP.

5.Undesirable outputs: The following were chosen for this indicator: (1) total wastewater discharge; and (2) COD discharge in wastewater. Relevant data were obtained from the China Statistical Yearbook [42].

Here, we selected five main factors—water resource endowment, technology development level, water pricing, government environmental control strength, and water resources utilization structure—as explanatory variables to analyze their impact on water resource green efficiency in the YREB. Relevant data were obtained from the China Statistical Yearbook [42].

(1) Water resource endowment (water): This is the amount of water resources per capita. The abundance of water resources and their utilization efficiency are closely related. Resource endowment generally has an inverse effect on the regional resource utilization efficiency.

(2) Level of scientific and technological development (tech): In general, scientific and technological progress can provide enterprises with technologies to save water and reduce wastewater discharge; further, science and technology can also promote wastewater treatment. Here, we measured this indicator using the proportion of GDP invested in the R&D of enterprises in various provinces and cities. Higher values indicate more robust technological innovation capacity, and vice versa.

(3) Water pricing (price): Existing studies consider water price to be an essential factor affecting regional water usage efficiency and water-saving efficiency. Due to the lack of systematic data on water prices in provinces and municipalities, the data on provincial capitals’ prices were selected as a proxy.

(4) The strength of the government’s environmental regulations (regu): The share of emissions fee revenues in the GDP of each province and city in the YREB was used to measure this indicator.

(5) Structure of water resource use (stru): Water use efficiency differs significantly based on the use of water. It is necessary to control for the effect of the structure of water use on the service efficiency. Here, we use the ratio of the sum of water use in agricultural and industrial production to the total water use.

### 2.2. Research Methodology

#### 2.2.1. SBM-DEA Model

We use the undesirable outputs SBM-DEA model proposed by Tone [47] for measuring the green efficiency of water resources in each YREB region. This model considers undesirable outputs to solve the inefficiency problem caused by slackness in the general radial DEA model. Furthermore, this method makes water resources and green efficiency comparable across decision making units (DMUs), thereby facilitating accurate measurement of each DMU’s water resource green efficiency. The SBM model for a specific DMU is as follows [48]:(1)$$\rho =\min\frac{1-\frac{1}{I}\sum_{i=1}^{I}s_{i}^{x}/x_{k^{\prime }i}^{t^{\prime }}}{1+\frac{1}{E+U}\left(\sum_{e=1}^{E}s_{e}^{y}/y_{k^{\prime }e}^{t^{\prime }}+\sum_{u=1}^{U}s_{u}^{z}/z_{k^{\prime }u}^{t^{\prime }}\right)}\;s.t.\;\sum_{t=1}^{T}\sum_{k=1}^{K}\lambda_{k}^{t}x_{ki}^{t}+s_{i}^{x}=x_{k^{i}}^{t^{\prime }},\;i=1,\cdots ,I\;\sum_{t=1}^{T}\sum_{k=1}^{K}\lambda_{k}^{t}y_{ke}^{t}-s_{e}^{y}=y_{k^{\prime }e}^{t^{\prime }},\;e=1,\cdots ,S_{1}\;\sum_{t=1}^{T}\sum_{k=1}^{K}\lambda_{k}^{t}z_{ku}^{t}+s_{u}^{z}=z_{k^{\prime }u}^{t^{\prime }},\;u=1,\cdots ,U\;\lambda_{k}^{t}⩾0,s_{i}^{x}⩾0,s_{e}^{y}⩾0,s_{u}^{z}⩾0,k=1,\cdots ,K$$
where ρ, I, E, U, and *K* are the green efficiency values of water resources to be calculated, the number of input types, the number of desirable output types, and the number of undesirable output types, respectively; x, y, and z denote the vectors of inputs, desirable outputs, and undesirable outputs, respectively; $s_{i}^{x}$ and $s_{u}^{z}$ represent the redundancy of inputs and undesirable outputs, respectively; $s_{e}^{y}$ is the deficit in desirable outputs; $\left(x_{k^{\prime }i}^{t\prime },y_{k^{\prime }e}^{t^{\prime }},b_{k^{\prime }u}^{t^{\prime }}\right)$ is the first $k^{\prime }$ of the production unit $t^{\prime }$ input-output values for the period; and $\lambda_{k}^{t}$ is the weight of the decision unit. The objective function of ρ with respect to $s_{i}^{x},s_{e}^{y},s_{u}^{z}$ strictly monotonically decreases, and 0<ρ≤1. When ρ=1 and $s_{i}^{x}=s_{e}^{y}=s_{u}^{z}=0$, the decision unit evaluated is efficient and there is no input–output redundancy or deficiency. When ρ<1, the evaluated decision unit has efficiency loss. Namely, DEA is ineffective, and one needs to improve the green efficiency of water resources by optimizing the number of inputs and outputs.

#### 2.2.2. Malmquist TFP Index Model

As the SBM-DEA model undertakes a static analysis of the green efficiency of water resources, this study constructs a Malmquist index to dynamically examine TFP. The Malmquist TFP index was introduced by Malmquist in 1953 [49]. Fare et al. [50] combined it with DEA to construct the Malmquist productivity change index of intertemporal variation $MI_{t}^{t+1}$. This index is used to objectively measure the relationship between technical efficiency change (EC), technological change (TC), and the TFP index ($MI_{t}^{t+1}$).
(2)$$MI_{t}^{t+1}={\left[\frac{D^{t}\left(x^{t+1},y^{t+1},z^{t+1}\right)}{D^{t}\left(x^{t},y^{t},z^{t}\right)}\times \frac{D^{t+1}\left(x^{t+1},y^{t+1},z^{t+1}\right)}{D^{t+1}\left(x^{t},y^{t},z^{t}\right)}\right]}^{\frac{1}{2}}$$

If $MI_{t}^{t+1}>1$, it indicates an increase in the TFP level. If $MI_{t}^{t+1}<1$, it indicates a decrease.

The $MI_{t}^{t+1}$ index can be decomposed into the EC and TC indices; if the constraint $\sum_{\kappa =1}^{K}\lambda_{\kappa }=1$ is used in the SBM model, it is transformed into the variable returns to scale (VRS) SBM model. The EC model can further be decomposed into pure technical efficiency change (PEC) and scale efficiency change (SEC). Thus, Equation (2) can be written as follows:(3)$$MI_{t}^{t+1}=\frac{P^{t}\left(x^{t},y^{t},z^{t}\right)}{P^{t}\left(x^{t+1},y^{t+1},z^{t+1}\right)}\times \frac{S^{t}\left(x^{t+1},y^{t+1},z^{t+1}\right)}{S^{t}\left(x^{t},y^{t},z^{t}\right)}\;\times {\left[\frac{D^{t}\left(x^{t+1},y^{t+1},z^{t+1}\right)}{D^{t+1}\left(x^{t+1},y^{t+1},z^{t+1}\right)}\times \frac{D^{t}\left(x^{t},y^{t},z^{t}\right)}{D^{t+1}\left(x^{t},y^{t},z^{t}\right)}\right]}^{\frac{1}{2}}$$
where $MI_{t}^{t+1}$ is the TFP index of the decision unit, representing the intertemporal dynamics of water resource green efficiency; $P^{t}$, $S^{t}$, and $D^{t}$ are the PEC, SEC, and TC, respectively. If the TFP and decomposition indices are greater than 1, the corresponding TFP and its decomposition show an upward trend, and vice versa.

#### 2.2.3. Social Network Analysis Model

We use the social network analysis model to study the spatial correlation network characteristics of the green efficiency of water resources in the YREB. Social network analysis is a useful tool for studying the relationships between actors and their quantifying connections. Therefore, it is crucial to identify the “relationships” before conducting network analysis [51]. We use the VAR Granger causality test [52] to determine the relationship between the green efficiency of water resources of each region. First, the time series of water resource green efficiency of two YREB regions is defined as $\left\{A_{t}\right\}$ and $\left\{B_{t}\right\}$. Second, two VAR models are constructed to test whether there is a Granger causality relationship between the changes in water resource green efficiency in the two regions.
(4)$$A_{t}=\alpha_{1}+\sum_{i=1}^{m}\beta_{1,i}a_{t-i}+\sum_{i=1}^{n}\gamma_{1,i}b_{t-i}+\varepsilon_{1,t}$$
(5)$$B_{t}=\alpha_{2}+\sum_{i=1}^{p}\beta_{2,i}a_{t-i}+\sum_{i=1}^{q}\gamma_{2,i}b_{t-i}+\varepsilon_{2,t}$$
where $a_{i},\beta_{i},\gamma_{i}(i=1,2)$ are the parameters to be estimated; $\left\{\varepsilon_{i,t}\right\}(i=1,2)$ is the residual term and obeys the standard normal distribution; and m, n, p, and q represent the lagged order of the autoregressive term. Suppose the test result is the Granger factor between regions *X* and *Y*. Then, there is a significant spatial association effect of region *X* on region *Y.* Accordingly, a directed line is drawn between the two regions from *X* to *Y*. Similarly, we develop a green efficiency spatial correlation network of water resources in the 11 YREB regions.

For the overall structural characteristics of this spatially linked network, four indicators are used to measure the magnitude of benefits, network robustness, asymmetric accessibility of nodes, and number of inter-node linkage channels: network density, network relatedness, network hierarchy, and network efficiency, respectively.

#### 2.2.4. Dynamic Panel System GMM Model

According to the calculation process of the undesirable output SBM-DEA model, any factor that affects the input and output of water resources will also impact the green efficiency of water resources in the YREB. To accurately measure the relationship between this green efficiency and the influencing factors, we construct a dynamic panel system GMM model by introducing the first order lagged terms of the explanatory variables into the static model [53]. Following the method proposed by Auffhammer and Richard [54], we develop the following dynamic panel model.
(6)$$\begin{array}{cc} ln\;effic_{i,t} & =\alpha_{i}+\lambda \;ln\;effic_{i,t-1}+\beta_{1}\ln water_{i,t}+\beta_{2}\ln tech_{i,t}+\beta_{3}\ln pric_{i.t}+\beta_{4}\ln regu_{i,t}\\ & +\beta_{5}\ln stru_{i,t}+\eta_{i}+\varepsilon_{i.t} \end{array}$$
where $\ln effic_{i,t}$, $\ln effic_{i,t-1}$, ${\ln\mathrm{water}}_{i,t}$, $\ln tech_{i,t}$, ${\ln\mathrm{pric}}_{i,t}$, ${\ln\mathrm{regu}}_{i,t}$, and $ln\;stru_{i,t}$ denote the logarithm of green efficiency, lagging period green efficiency, water resource endowment, level of scientific and technological development, water price, government environmental control efforts, and water resources utilization structure, respectively. α is a constant, $\beta_{i}$ are the regression coefficients of the explanatory variables, $\eta_{i}$ are the inter-provincial individual effects, and $\varepsilon_{i,t}$ represents the random error.

Two problems may arise in estimating this model: the presence of individual effects across provinces in the regression model, and possible endogeneity between the explained and explanatory variables. The endogeneity problem leads to biased and inconsistent estimates of both fixed and random effects; Arellano and Bond [55] proposed a GMM estimation method to address these problems, including two forms of differential GMM and a systematic one. Compared to the former, the system GMM estimation method can estimate the coefficients of variables that do not vary over time and that are not prone to the weak instrumental variable. This leads to improved estimation efficiency. Thus, this study adopts the system GMM estimation method.

## 3. Empirical Results and Analysis

### 3.1. Results of Green Efficiency Measurements of Water Resources in the YREB

We use the SBM-DEA model with undesirable outputs to estimate the green efficiency of water resources in each YREB region. The specific calculation results are presented in Table 2. In the SBM-DEA model, to measure water resource green efficiency, we set the weight ratio of desirable outputs to undesirable outputs as 1:1. Then, we measure the green efficiency of water resources containing undesirable outputs.

Temporally, the green efficiency value from 2002 to 2017 fluctuates and shows a general decreasing trend (see Table 2). However, there are both positive and negative changes within this period. For example, the water resource green efficiency improved year on year between 2002 and 2010, rising by 12.95% in total. It reached a peak of 0.977 in 2010. This was because of the strict management of water resource use by the state, industrial transformation, upgrading, and the continuous development of water-saving technologies. However, the efficiency began falling thereafter, declining to 0.697 in 2017. Across the entire period, the average value of water resource green efficiency in the YREB was 0.859. This indicates that the efficiency in the YREB was generally high, and the goal of synergistic economic, social, and environmental development was initially achieved.

Regionally, the green efficiency of water resources in the YREB shows noticeable regional differences (see Figure 2). The efficiency in the downstream regions is the highest, followed by the upstream regions. The utilization of midstream regions is not optimistic. This indicates that the water resource utilization efficiency, based on the concept of green development, does not show a positive correlation with economic growth in this region. Water resource utilization efficiency is generally high in the downstream regions, with efficiency values ranging from 0.900 to 1.000. For example, from 2005 to 2013, the green efficiency of water resources was 1.000, which is the production frontier. The water resource efficiency in the midstream regions fluctuates between 0.600 and 1.000. However, this value is at a lower level among the upper and middle distances. Furthermore, the efficiency decreased to 0.640 in 2017. The upstream regions also experience fluctuations in efficiency. From 2002 to 2008, efficiency declined slightly and then increased. In 2009 and 2010, the efficiency value was 1.000. However, since 2011, efficiency began declining and decreased to 0.759 in 2017.

As shown in Figure 3, during the study period, the YREB regions with the highest green efficiency of water resources are Shanghai, Chongqing, Jiangxi, and Guizhou. The average efficiency value in these areas was equal to 1, while the rest were inefficient areas. This indicates that the overall green efficiency of water resources in the YREB is low. The input-output ratio must be changed to improve the green efficiency of water resources to make these resources useful.

### 3.2. Analysis of the Green Efficiency Water Resources in the YREB Based on the Malmquist Index

To further analyze the dynamic trend of water resource green efficiency in the YREB, this study measures the TFP and its decomposition index based on the Malmquist TFP index model. The results are shown in Table 3. The mean values of TFP change and its decomposition index by year and province are shown in Figure 4.

Regarding the Malmquist Index (MI), the average value during the study period was 1.007, with an increase of 0.70%. The relevant inter-period dynamic changes are small, indicating that under resource and environmental constraints, the overall green efficiency of water resources in the YREB is on an upward trend while fluctuating between years. Figure 3 intuitively shows that the change in TFP is closely related to the EC index. Therefore, the EC index is one of the critical factors affecting green water resource development.The mean value of the PEC index from 2002 to 2017 was 0.990, a decrease of 1.00%. Further, the mean value of the SEC index was 1.000. This indicates that during the study period, pure technical efficiency regressed slightly while scale efficiency remained unchanged.In terms of the TC index, the average value was 1.016 over the 16 years, which indicates that China’s technological level has continuously improved over the past 16 years. The increasingly severe water crisis prompted the 11 YREB regions to pay more attention to advanced technology investment in water resource utilization.Overall, the improvement in the TFP of the green efficiency of water resources in the 11 YREB regions relies mainly on technological changes. By contrast, the changes in pure technical efficiency and scale efficiency restrict the improvements in efficiency to a certain extent. This indicates that the advancements in efficiency are due to scientific and technological progress. Conversely, the government’s regulation and control measures, management tools, and the expansion of the overall scale of water resources use, to a certain extent, hinder the improvement of the green efficiency of water resources. Thus, these must be optimized.

### 3.3. Overall Structural Characteristics of the Spatially Linked Network of the Green Efficiency of Water Resources in the YREB

We use the VAR Granger causality test to examine network linkages between the water resources and green efficiency data of the 11 YREB regions. Based on the constructed spatial correlation relationship matrix of water resource green efficiency in the YREB, the spatial correlation network of 11 YREB regions is drawn using the visualization tool Netdraw and Ucinet software (Figure 5).

This spatial correlation network takes the 11 YREB regions as nodes. According to the Granger causality test results, the total number of relationships is 34. The overall network density is 0.6182, indicating that the water resource green efficiency in the YREB regions shows significant spatial correlation relationships. Furthermore, the degree of closeness is high in terms of values (the higher the density, the closer the connection). Provinces and cities need to strengthen the spatial linkage of water resource green efficiency to enhance the stability of the spatial correlation network. The network correlation is 1, which indicates that all 11 YREB regions are connected to each other in terms of water resources and green efficiency development. No region is independent of the spatial correlation network, and the network nodes are well connected. The network efficiency is 0.4667, indicating fewer redundant connections in the network. This reflects more spatial spillover paths of water resource green efficiency in the YREB and better network stability.

### 3.4. Analysis of Spatial and Temporal Differences in the Green Efficiency of Water Resources in the YREB

Figure 6 shows the spatial divergence of green water resource efficiency in the YREB in 2002, 2007, 2014, and 2017. According to the empirical results, the efficiency values are divided into five interval segments. Higher efficiency values are denoted by darker colors. The efficiency changed significantly from 2002 to 2017 and showed a decreasing trend. The provinces and cities with deeper colors in 2002 were Shanghai, Anhui, Zhejiang, Jiangxi, Chongqing, Guizhou, and Yunnan. The provinces and cities with deeper colors in 2017 were Jiangsu, Shanghai, Jiangxi, Chongqing, and Guizhou. Anhui, Zhejiang, Hunan, and Yunnan showed decreasing efficiency values to a greater degree. Only one place, Jiangsu, showed an increase in the efficiency values.

From a regional perspective, the downstream regions (Shanghai, Jiangsu, Zhejiang, and Anhui) have higher economic development levels, social construction, and regional development than the midstream and upstream regions. The high degree of industrially developed chains, large industrial effluent discharges, and the lack of water-saving technologies have led to a decreased value of water resource green efficiency.

The midstream regions (Jiangxi, Hubei, and Hunan) have relatively low water resource utilization efficiency and show no signs of rising. The midstream region is a traditional, old industrial base that has been supported by the state for a long time and includes resource-based industries such as automobile and parts manufacturing, and the mineral industry. Industrial pollution discharge, old technology, and an overdependence on resources have perhaps brought about a “resource curse” in these regions. The issue of water resource pollution here is serious, especially coupled with the lack of awareness of environmental governance. The midstream and downstream regions should accelerate transformation and upgrades, guide the development of industrial clusters, and develop and expand energy-saving and environmental protection industries. Further, improving R&D of environmental protection technologies, focusing on controlling the total amount of industrial water consumption, and realizing manufacturing industries’ greening are other important points.

The upstream YREB is uniquely endowed with natural resources, accounting for 48% of the total water resources of the YREB. The upstream YREB performs a crucial function as a green barrier, building the YREB’s ecological base and performing essential tasks to protect biodiversity, such as conserving the water of China’s major rivers. With the launch of major national strategies such as Western Development, the YREB, the ecological protection and high-quality development of the Yellow River Basin, and the “One Belt, One Road” initiative, the upstream regions have become important areas for the westward inflow and the embracing of capital, technology, and labor. The impacts of industry and population have reduced the carrying capacity of water resources in these regions, and thereby reduced the green efficiency of water resources. The upstream regions should adhere to the path of putting ecology first and fostering green development to ensure that the economy and the environment develop synergistically.

### 3.5. Analysis of Factors Affecting the Green Efficiency of Water Resources in the YREB

We conducted a dynamic panel system GMM regression analysis of water use efficiency and its influencing factors in 11 YREB regions from 2002 to 2017. The results are shown in Table 4. The residual terms of the model satisfy the first-order serial correlation and are second-order serially uncorrelated. This indicates that the model is reasonably constructed. The *p*-value of Hansen’s test is 1.0000, which is greater than 0.05. This means that the use of the model’s instrumental variables is valid, and there is no overidentification problem.

The regression results show that the green efficiency of water resources in the previous period positively affects green efficiency in the current period, with a regression coefficient of 0.7626 at the 1% significance level. This indicates that it is reasonable to introduce a first-order lag of the explanatory variable in the regression model.The increased per capita water resources hurt water resource green efficiency at the 1% significance level. For every unit increase in per capita water resources, water resource utilization efficiency decreases by 0.0497. The abundance of water resources affects the awareness of water conservation among residents of the region. In regions with more abundant water resources, people have insufficient awareness of water conservation and pay little attention to the development and utilization of water resources. At the same time, abundant water resources have a negative correlation to the demand for relevant technologies. In provinces where water resources are relatively scarce, people have a strong awareness of water conservation, and there is less wasteful and predatory water resource exploitation. Thus, there is a negative correlation between per capita water resources and the water usage efficiency.The higher the level of regional technological development, the higher the local water purification capacity, water supply capacity, and water productivity. However, the estimation results show that the effect of the level of technological development is minimal and not statistically significant. Possible reasons for this include a low level of science and technology in the provinces of the YREB, especially the lack of cutting-edge core technologies for water pollution treatment, and the high cost of science and technology R&D. By contrast, with a weak ability to transform scientific and technological achievements, the corresponding technology cannot be applied practically due to the high cost and low rate of scientific and technological inputs and outputs.Water price is an economic lever to regulate water resource supply and demand. It is a market signal that reflects water resource scarcity and is the “invisible hand” behind the rational allocation of water resources. Table 4 shows that the water price is negatively related to the green efficiency of water resources. This indicates that China’s current water pricing system does not improve water resource utilization efficiency. The unreasonable water pricing policies and mechanisms are among the main reasons for the shortage and wastage of water resources in China. The low water prices that have been set by government departments for a long time have made it difficult to reverse the situation, where the total demand exceeds the total supply, resulting in low green efficiency of water resources.There is a significant positive correlation between the strength of government environmental regulation and the green efficiency of water resources, indicating the presence of a “visible hand.” By increasing environmental regulations, the government can create incentives for enterprises to save water and reduce emissions. This can help improve water resource efficiency. The YREB regions have greatly improved the green efficiency of water resources by establishing a comprehensive and strict sewage charging system.There is a significant negative correlation between water use in both agriculture and industrial production, and the green efficiency of water resources. The higher the proportion of agricultural and industrial water, the lower the green efficiency of water resources. This indicates that the redundant water inputs in China’s agricultural and industrial production have not been fundamentally reversed, and water use is still relatively sloppy. Agriculture is a significant water user. The low efficiency of agricultural water use is mainly reflected in inefficient irrigation methods. Simultaneously, most Chinese industries are high-energy and water-consuming industries, which use many water resources and emit many pollutants. This results in the low efficiency of industrial water use in general.

## 4. Conclusions

The undesirable outputs SBM-DEA, Malmquist index, and social network analysis models were used to measure the green efficiency of water resources in each YREB region and analyze the spatial correlations across regions. The dynamic panel system GMM model was used to analyze the factors affecting this green efficiency.

The results revealed:Temporally, the green efficiency of water resources in the YREB fluctuates while showing a decreasing trend; spatially, the efficiency shows a significant spatial non-equilibrium and decreases in the following order: downstream, upstream, and midstream.The change in the TFP index of the YREB regions shows an overall upward trend. Pure technical efficiency regresses slightly, while scale efficiency remains unchanged. The improvement in TFP mainly relies on technical changes.The green efficiency of water resources in the YREB shows a significant spatial correlation. All regions are in the spatial correlation network, which is very stable.The GMM regression results show that water endowment, water price, government environmental control strength, and water utilization structure significantly affect the water resource green efficiency in the YREB.

We propose some suggestions for ensuring the gradual improvement of the green efficiency of water resources in the YREB, and achieving ecologically sound green development:Optimize the industrial layout and narrow the regional differences in the green efficiency of water resources in the YREB: (a) Optimize the regional and riverine industrial layout, including the upstream, midstream, and downstream regions, (b) promote the implementation of differential environmental access upstream and downstream of the basin, developing a regional ecological linkage mechanism based on the golden waterway for the YREB’s ecological development and for forming a pattern of complementary advantages and collaborative interactions between the upstream, midstream, and downstream; and (c) narrow the development gap between upstream, midstream, and downstream regions;(a) Accelerate technological innovation, increasing investment in science and technology and cultivating scientific and technological talents to provide technical support for the improvement of green efficiency of water resources in the YREB, (b) implement innovation-driven and green technology enhancement in the YREB, promoting strategies that prioritize water conservation and strengthening the construction of technical systems and experimental base platforms for efficient water resource development and utilization;Improve the development of a water rights market system for water resources in the YREB, creating a novel water resource price management system, promoting the rational allocation and economical use of water resources, maintaining a virtuous cycle of water resources, and safeguarding and improving the green efficiency of water resources; and(a) Effectively use price leverage to strengthen and enhance water pollution prevention and control, (b) improve the sewage treatment mechanism in the YREB, and (c) guide the optimal allocation of resources, realizing the internalization of ecological and environmental costs and promoting the green development of the YREB and ecological civilization.

## Figures and Tables

**Figure 1 ijerph-18-03101-f001:**
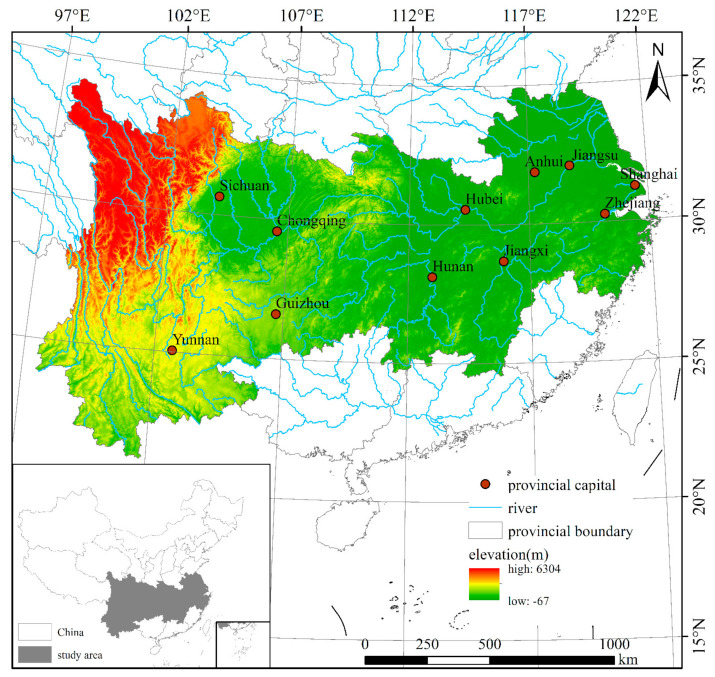
The Yangtze River Economic Belt in China.

**Figure 2 ijerph-18-03101-f002:**
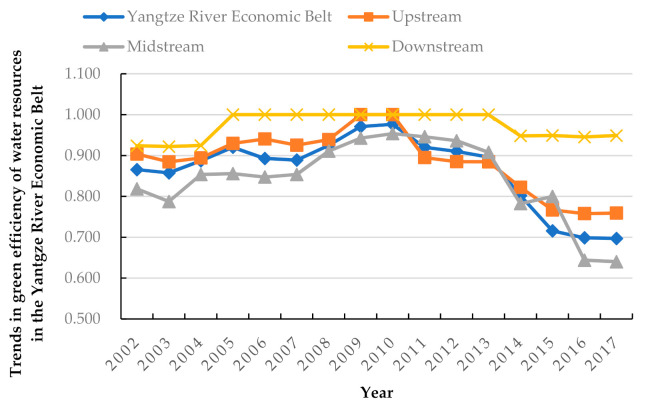
Trends in green efficiency of water resources in the Yangtze River Economic Belt, and its upstream, midstream, and downstream regions from 2002–2017.

**Figure 3 ijerph-18-03101-f003:**
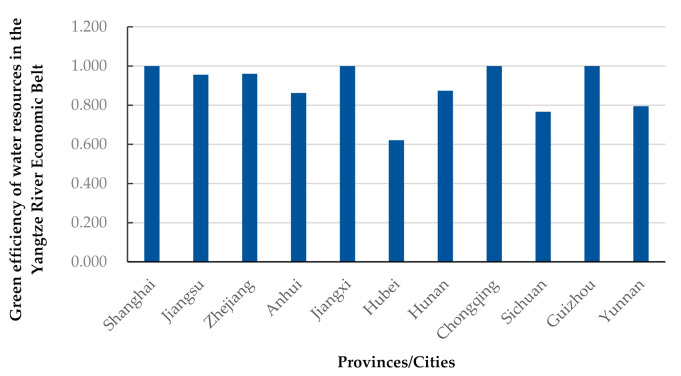
Green efficiency of water resources in 11 provinces and cities in the Yangtze River Economic Belt.

**Figure 4 ijerph-18-03101-f004:**
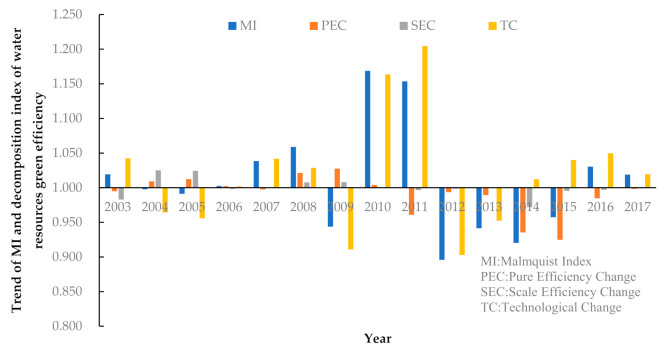
Trend of MI and decomposition index of water resource green efficiency in Yangtze River Economic Belt.

**Figure 5 ijerph-18-03101-f005:**
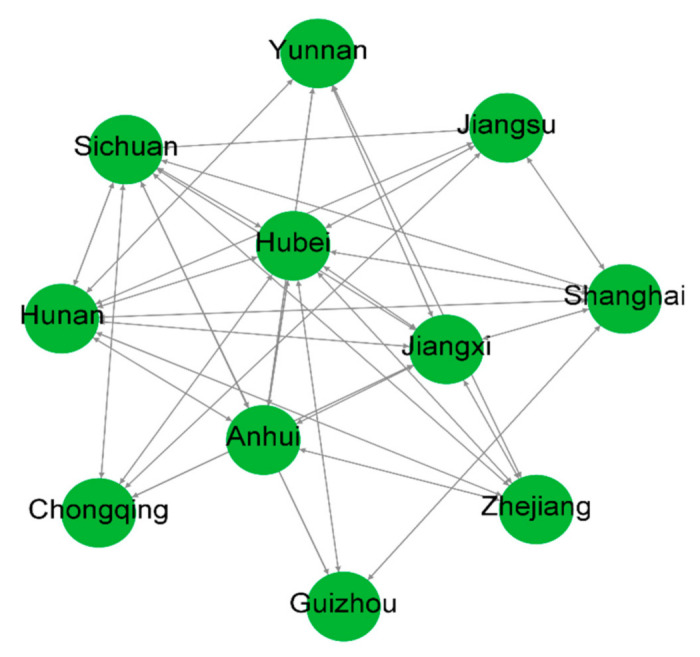
Spatial correlation network of green efficiency of water resources in the Yangtze River Economic Belt.

**Figure 6 ijerph-18-03101-f006:**
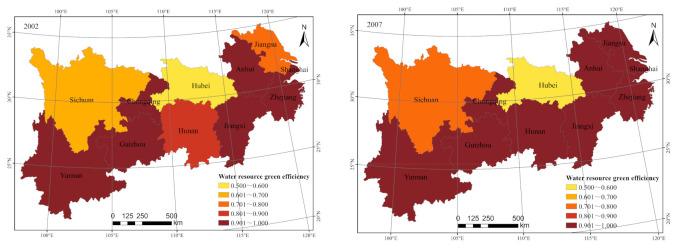
Spatial-temporal evolution of green efficiency of water resources in the Yangtze River Economic Belt.

**Table 1 ijerph-18-03101-t001:** China Human Development Index (CHDI) indicator system.

Level I Indicators	Secondary Indicators	Tertiary Indicators
China Human Development Index (CHDI)	Life expectancy index	Average life expectancy
	Education index	Average years of schooling
	Income index	Gross national income per capita
	People’s livelihood improvement index	Engel’s coefficient
		Social security index
	Sustainable development index	Innovation development index
		Green development index
		Openness index

**Table 2 ijerph-18-03101-t002:** Green efficiency of water resources in the Yangtze River Economic Belt as a whole, and by region, for 2002–2017.

District	2002	2003	2004	2005	2006	2007	2008	2009	2010	2011	2012	2013	2014	2015	2016	2017	Average Value
Yangtze River Economic Belt	0.865	0.858	0.887	0.920	0.893	0.889	0.925	0.971	0.977	0.920	0.910	0.896	0.802	0.716	0.699	0.697	0.859
Upstream	0.904	0.885	0.894	0.930	0.940	0.925	0.939	1.000	1.000	0.895	0.885	0.885	0.822	0.767	0.758	0.759	0.884
Midstream	0.818	0.787	0.854	0.856	0.847	0.854	0.911	0.943	0.954	0.946	0.936	0.908	0.782	0.800	0.644	0.640	0.837
Downstream	0.924	0.922	0.925	1.000	1.000	1.000	1.000	1.000	1.000	1.000	1.000	1.000	0.948	0.949	0.945	0.949	0.972

**Table 3 ijerph-18-03101-t003:** Mean values of Malmquist Index (MI) and its decomposition index of green efficiency of water resources in the Yangtze River Economic Belt.

Region	EC	TC	PEC	SEC	MI
Shanghai	1.000	1.000	1.000	1.000	1.000
Jiangsu	1.016	1.058	1.000	1.016	1.075
Zhejiang	0.990	1.052	1.000	0.990	1.041
Anhui	0.956	1.043	0.960	0.997	0.997
Jiangxi	1.000	0.981	1.000	1.000	0.981
Hubei	1.007	1.003	1.003	1.005	1.011
Hunan	0.972	1.039	0.966	1.006	1.010
Chongqing	1.000	1.000	1.000	1.000	1.000
Sichuan	0.993	1.028	0.996	0.997	1.020
Guizhou	1.000	0.989	1.000	1.000	0.989
Yunnan	0.961	0.992	0.968	0.993	0.954
Average value	0.990	1.016	0.990	1.000	1.007

**Table 4 ijerph-18-03101-t004:** Estimation results of the dynamic panel system generalized method of moments (GMM) model.

Variable	Coefficient	Std. Error	z-Statistic	Prob
*L. lneffi*	0.7626 ***	0.0531	14.36	0.000
*lnwater*	–0.0497 ***	0.0193	–2.58	0.010
*lntech*	0.0036	0.0052	0.69	0.489
*lnpric*	–0.0769 *	0.0403	–1.91	0.057
*lnregu*	0.0102 *	0.0052	1.93	0.053
*lnstru*	–0.4140 *	0.2320	–1.78	0.074
*_cons*	0.4972 ***	0.1286	3.87	0.000
*N*	165			
AR(1) (*p*-value)	0.0413			
AR(2) (*p*-value)	0.2634			
Hansen test (*p*-value)	1.0000			

Standard errors in parentheses, * *p* < 0.1, ** *p* < 0.05, and *** *p* < 0.01.

## Data Availability

The data presented in this study are available on request from the corresponding author. The data are not publicly available due to legal and privacy issues.

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
