# Peer review of "Spatial and Temporal Differences in the Green Efficiency of Water Resources in the Yangtze River Economic Belt and Their Influencing Factors"

_ijerph, 2021, doi:10.3390/ijerph18063101_

Round 1

Reviewer 1 Report

1. The format of the whole manuscript, especially the Reference, does not conform to the format of IJERPH. The authors should revised the whole paper according to the instructions and the template of the journal carefully. Moreover, there should be more references from IJERPH. 2. Although the authors used several methods in the “Research methods and data” section, they did not clearly state the relevance of these methods when studying the subject of this paper, and the innovations brought about by the above-mentioned relevance. The authors should elaborate it. 3. In Line 140-188, the format is incorrect. 4. In Line 290-291, the title format is incorrect. 5. The latest literature about topic of DEA and Green Efficiency should be cited to explain the topic of this research [1-2]. [1] Efficiency evaluation of industrial waste gas control in China: A study based on data envelopment analysis (DEA) model. Journal of Cleaner Production 2018, 179, 1-11. [2] Analysis of the Effectiveness of Air Pollution Control Policies based on Historical Evaluation and Deep Learning Forecast: A Case Study of Chengdu-Chongqing Region in China. Sustainability 2021, 13(1), 206.

Reviewer 2 Report

Thank you for giving me the opportunity to review the paper entitled “Analysis of Spatial and Temporal Differences in Green Efficiency of Water Resources in the Yangtze River Economic Belt 3 and the Influencing Factors”. The authors set out to analyze regional differences and spatial characteristics of green efficiency of water resources using the SBM-DEA undesirable output model, Malmquist index and social network analysis, respectively, and established a dynamic panel model to examine empirically the main factors that influence the green efficiency of water resources along the Yangtze River Economic Belt.

Some comments:

The paper has several problems:

  1. English has to be grammatically revised and the authors will have to choose between US English and UK English (labor / labour);
  2. the methodologies must be presented in a little more detail;
  3. the authors must always adopt the same designation for methodologies! There is no coherence ... for example: non-consensual output-oriented SBM-DEA model, non-expected output-oriented SBM-DEA model, non-desired output-oriented SBM-DEA model. In the case of this methodology, authors should read and refer:

Tone, K.. “Dealing with Undesirable Outputs in DEA: A Slacks-based Measure (SBM) Approach.” (2003).

  1. The Malmquist Total Factor Productivity Index model must be revised and better explained.
  2. Please do not confuse efficiency with effectiveness.
  3. In data section, what percentage of data is missing?
  4. the calculation of the China Social Development Index (CHDI) for the years being studied is quite questionable!!!
  5. Table 1 is not is not referred to in the text
  6. the name of section 3 is on the line 290!!!
  7. where can we see this? “thanks to the strict management of water resources use by the state, industrial transformation, and upgrading, and the continuous development of water-saving technologies.”
  8. The sentence: The middle reaches' utilization is not optimistic, indicating that the efficiency of water resources utilization based on the concept of green development does not show a positive correlation with economic” Must be clarified at this point
  9. The sentence that starts in line 357 and ends in line 365 is very long and difficult to read.
  10. Arellano et al. isn´t in the references
  11. the text needs a stronger link between parties with different methodologies
  12. There are some typos

Reviewer 3 Report

Dear Authors,

The article presents a spatial analysis of the green efficiency of water resources. The unexpected output SBM-DEA model and the Malmquist index, which are classic in this type of analysis, were used. The analyzes carried out on the basis of the Yangtze River Economic Belt are local studies, but would certainly be more valuable if authors refer to other studies  and conducting a discussion. I wonder why, for example, the work of Guo et al. 2020 "Valuation of ecological use of water resources based on the SBM-TOBIT panel model: case study from Henan Province, China" not included? This analysis of the work has similar problems. Overall, work is prepared properly besides a few comments. In my opinion problem raised is presented properly but discussion must be included additional. After taking into account my suggestions, the work is recommended for publication.

Reviewer 4 Report

  1. Please explain the abbreviations used in the paper, e.g. GDP (line 73), SBM-DEA model (line 82), EBM (line 89), GMM model (line 114).
  2. There is no description of the object (chapter 2). Please write which region was chosen for the research, what provinces it covers, what is its area, number of inhabitants, etc.
  3. Please complete the description of Yangtze River Economic Belt of a drawing or a map that shows the mentioned regions (lines 40-54).
  4. Line 127 – Perhaps it would be better to use Simple Past to describe the research methodology.
  5. Please add the literature quotations (e.g. lines 112, 155, 179 (VAR Grager causality test), -  lines 212 - 229 – China statistical Yearbook, China Water Resources Bulletin; China Population and Employment Yearbook, China Social Development Index; lines 262-290 (data sources should be provided). Lines 79, 96, 127 - no literature sources are given.
  6. There is no reference in the text to the table 1.
  7. In figures 1-3, the axes should be described.
  8. In Figure 5 the descriptions are illegible
  9. Point 3.5.1. is this not a methodology description?
  10. The conclusion must be rewritten. It is more of a summary and therefore it is much too long.

Round 2

Reviewer 1 Report

Authors have addressed the comments in the revised version of manuscript. The current version is appropriate for publication.

Author Response

Thank you very much for your positive comments.

Reviewer 2 Report

This version is much better and all the flaws have been fixed. The suggestions were answered.
Congratulations on the final product.

Author Response

(The authors gave the same response as above.)
